# ACCELERATION-AWARE SAMPLING FOR FEW-STEP RECTIFIED FLOW MODELS

## ABSTRACT

Rectified flows (RFs) enable efficient, high-fidelity image synthesis by integrating a learned velocity field from noise to data. However, in latency-constrained scenarios with few-step settings, RFs tend to produce degraded images. We identify this limitation in the commonly used Euler sampler which assumes piecewise-constant velocity and neglects inherent acceleration. Consequently, significant discretization errors occur and dominate the few-step sampling process. To address this issue, we introduce **A**cceleration-**A**ware **S**ampling (**A²S**) which explicitly accounts for acceleration while maintaining the same computational cost as Euler sampler. From a second-order perspective, we decomposes acceleration into temporal and spatial components, and compensates for both with lightweight approximations. Specifically, temporal acceleration is handled by a time-shifted velocity evaluation, aligning updates with mid-interval dynamics while preserving one forward pass per step. Meanwhile, spatial acceleration is captured by a smooth, time-dependent gain that modulates step size. As a result, A²S is model-agnostic, plug-and-play for existing pretrained rectified-flow models and requires no retraining. Across multiple models and benchmarks, A²S consistently improves image quality and stability in the few-step setting and remains competitive as the step count increases. Moreover, on FLUX, few-step A²S even surpasses standard multi-step samplers in image aesthetics and text–image alignment.

## 1 INTRODUCTION

Continuous-time flows have emerged as a powerful paradigm for high-fidelity image synthesis. In particular, rectified flows (RFs) (Lipman et al.; Liu et al.; Esser et al., 2024; Labs, 2024) provide a deterministic alternative to diffusion models (Ho et al., 2020; Rombach et al., 2022; Podell et al.) by learning a velocity field that transports noise to data. During inference, sampling proceeds by integrating velocity field from an initial noise to a final image. Though RF samplers are already more efficient than stochastic approaches, real-world applications such as interactive generation (Yu et al., 2024; 2025; Liang et al., 2025) and latency-sensitive services (Ma et al., 2025; Shen et al., 2025) often require strict time budgets, which force ultra-few sampling steps and expose a pronounced trade-off between quality and speed: when the number of integration steps get significantly reduced, image artifacts and loss of fidelity emerge (Lu et al., 2022).

We identify a central cause of this degradation in the Euler sampler (Lee et al., 2025; Esser et al., 2024) commonly used for RFs. Euler integration assumes that the velocity is piecewise constant within each step, i.e., zero acceleration. However, the learned velocity $v_\theta(x_t, t)$ in RFs depends on both the state $x_t$ and time $t$, producing trajectories with nonzero acceleration that Euler sampler neglects. In the few-step scenario, since the discretization interval is large, ignoring acceleration introduces significant integration error. This error accumulates across steps and becomes the dominant failure mode under few-step sampling.

To address this issue, we propose **A**cceleration-**A**ware **S**ampling (**A²S**) which explicitly incorporates the acceleration at the same computational cost. Specifically, A²S starts from a second-order Taylor view of the trajectory and decomposes the total acceleration into two components: (1) **Temporal acceleration** ($\mathbf{Acc_{temp}}$), which reflects how velocity varies with time at a fixed state; and (2) **Spatial acceleration** ($\mathbf{Acc_{spat}}$), describing how the velocity changes as the state moves along the trajectory.

Directly computing these acceleration terms is nontrivial, as it requires Jacobian-vector prod (JVP) (Blondel & Roulet, 2024) or higher-order differentiation, which is impractical for high-dimension image space. We compensates for them with lightweight approximations. For $\mathbf{Acc_{temp}}$, we uses a time-shifted evaluation: it evaluates the velocity as a shifted time within each step to better align the update with mid-interval dynamics, while keeping a single forward pass per step. For $\mathbf{Acc_{spat}}$, we approximate the effect of speed changes along the current direction of motion with a smooth, time-dependent gain that scales the step size. Technically, we utilize a simple linear scheduler to capture the dominant effect without additional model calls.

Across multiple pretrained RF models (Labs, 2024; Esser et al., 2024) and benchmarks (Lin et al., 2014; Wang et al., 2023; Kirstain et al., 2023; Wu et al., 2023), $A^2S$ consistently improves image quality and stability in the few-step setting and remains competitive as the number of steps increases. Notably, on FLUX, few-step $A^2S$ even surpasses standard multi-step samplers in terms of image aesthetics and text–image alignment, demonstrating that acceleration-aware integration can outperform naive increases in step count. Ablation studies further confirm that both temporal and spatial components contribute to the gains, and qualitative analyses show that $A^2S$ better tracks the underlying continuous-time flow. Our contributions are summarized as follows:

- We diagnose few-step degradation in rectified flows as a consequence of the zero-acceleration assumption in first-order samplers and derive a second-order view of RF sampling that explicitly separates temporal and spatial acceleration.
- We propose Acceleration-Aware Sampler that compensates temporal acceleration via a single time shift and approximates spatial acceleration through lightweight, time-dependent scale modulation.
- Our method exhibits significant quality gains in few-step setting across pretrained RF models and benchmarks, with minimal implementation overhead and no retraining; on FLUX-dev, few-step $A^2S$ surpasses standard multi-step samplers in both image aesthetics and text–image alignment.

## 2 RELATED WORKS

We review three main lines of work for few-step sampling: (i) reducing discretization error with more accurate numerical solvers to facilitate few-step sampling (Fast samplers), (ii) eliminating redundant sampling steps by caching and reusing past computations (Cache-based methods), and (iii) reduing the required number of sampling steps by knowledge distillation (Step distillation).

**Fast Samplers.** Fast samplers develop higher-order ODE solvers and timestep schedules to achieve strong fidelity with very few function evaluations. Representative examples include DPM-Solver and DPM-Solver++ (Lu et al., 2022; 2025), exponential-integrator formulations such as DEIS (Zhang & Chen), and unified predictor-corrector (UniPC) (Zhao et al., 2023). Schedule design (e.g., noise or sigma parametrization and nonuniform time grids) (Karras et al., 2022; Song et al.) is also crucial in the few-step setting. DEQ (Pokle et al., 2022) extends the DDIM solve for a joint fixed point with implicit differentiation, ParaDiGMS (Shih et al., 2023) uses Picard iterations within a sliding window to update multiple states simultaneously, ParaTAA (Walker & Ni, 2011; Shih et al., 2023) solves triangular nonlinear systems and introduces Triangular Anderson Acceleration for stable iterative refinement, and StreamDiffusion (Kodaira et al., 2023) combines batched denoising, residual classifier-free guidance, and asynchronous queues for real-time interaction. While these solvers can attain high order with 2–3 evaluations per step, extremely coarse discretizations (e.g., 4–8 steps) stress their smoothness assumptions.

**Cache-based Methods.** Cache-based strategy reuses past model evaluations to construct updates with a single function evaluation per step after initialization. DeepCache (Ma et al., 2024) identifies redundancy across nearby steps and reuses features to avoid recomputation while preserving fidelity at typical step counts. $\Delta$-DiT (Chen et al., 2024) optimizes diffusion transformers via step-sensitive block caching and adaptive allocation of computational resources. FORA (Selvaraju et al., 2024) reduces redundant inference by caching residuals between attention layers. TeaCache (Liu et al., 2025) exploits correlations between inputs and outputs to define dynamic caching policies. These methods are highly efficient as cached outputs substitute for extra forward passes. However, they do not explicitly address integration errors that arise under very coarse step sizes.

**Step Distillations.** Distillation reduces the number of network evaluations by training students to match a teacher's trajectory in fewer steps. Progressive distillation (Salimans & Ho) halves the

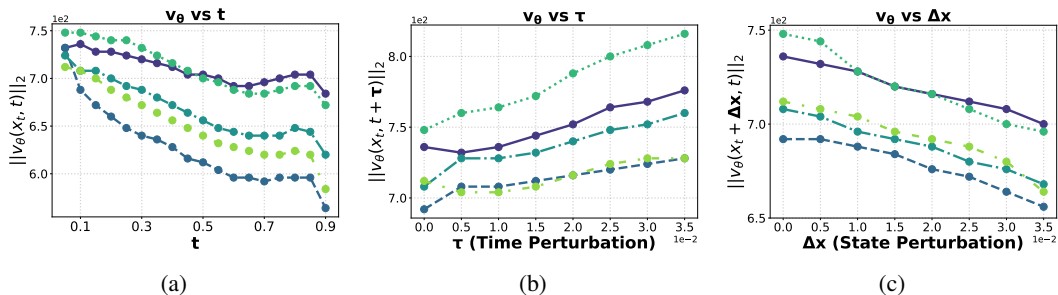

Figure 1: Empirical evidence that the learned velocity $v_\theta$ is not piecewise constant. (a) Velocity-norm $||v_\theta(x_t, t)||_2$ along the sampling trajectory for five samples. (b) Temporal sensitivity: $||v_\theta(x_t, t + \tau)||_2$ versus time perturbation $\tau$. (c) Spatial sensitivity: $||v_\theta(x_t + \Delta x, t)||_2$ versus state perturbation $\Delta x$.

step count iteratively, eventually reaching very small numbers of steps. Consistency-based approaches (Luo et al., 2023; Song et al., 2023) enforce self-consistency constraints to enable one-to-few step synthesis, including latent variants built on top of latent diffusion backbones. In the flow-matching and rectified flow literature, distillation and scaling studies (Liu et al.; Kim et al.; Lee et al., 2024) demonstrate straighter, more easily integrable trajectories and improved few-step quality; adversarial distillation (Sauer et al., 2024) further enhances perceptual fidelity at low step counts. These techniques require additional training or fine-tuning and may involve changes to the model architecture or loss.

## 3 METHOD

This work targets the few-step generation configuration, where standard Euler sampling in rectified flows often degrades image quality due to the ignorance of acceleration. We address this by explicitly modeling and compensating for both temporal and spatial components of acceleration within each integration step. We begin by analyzing the limitations of existing samplers that neglect acceleration (Sec. 3.1). Sec. 3.2 introduces $\mathbf{A^2S}$, which decomposes acceleration into temporal and spatial terms via a principled Taylor expansion. However, computing these accelerations is non-trivial, as either JVP or differentiation is required, which is almost intractable for high-dimension images. Subsequent subsections (Sec. 3.3 and Sec. 3.4) detail our efficient approximations for these components, enabling accurate compensation with minimal additional computation.

### 3.1 EMPIRICAL ANALYSIS OF NON-ZERO ACCELERATION IN RECTIFIED FLOW

**Preliminary.** We provide a concise introduction to rectified flow models (Lipman et al.; Liu et al.), covering their training and sampling procedures. RF learns a vector field $v_\theta$ by regressing the network output to a known velocity along a straightened (linear) probability path connecting a noise sample $x_0 \sim \mathcal{N}(0, I)$ and a data sample $x_1 \sim p_{data}$:

$$x_t = (1 - t)x_0 + tx_1, \quad t \in [0, 1] \tag{1}$$

This path has constant velocity $v = \frac{dx_t}{dt} = x_1 - x_0$. The model parameterized by $\theta$ is trained to predict this velocity by minimizing the mean-squared error:

$$\mathcal{L}_{\text{RF}}(\theta) = \mathbb{E}_{t,x_0,x_1}||v_\theta(x_t, t) - v||_2^2 \tag{2}$$

For sampling, time is discretized into $N$ steps with increment $\Delta t = 1/\text{N}$, and the learned velocity is integrated via an explicit iterative sampling process, progressing from $t = 0$ to $t = 1$:

$$x_{t+\Delta t} = x_t + v_\theta(x_t, t) \cdot \Delta t \tag{3}$$

**Motivation: Acceleration in Velocity Fields.** Sampling in RF models typically uses the first-order Euler integration in Eq. 3, which assumes $v_\theta$ is constant over each interval $[t, t + \Delta t]$. That is to say, the acceleration within each step is zero. We test this assumption empirically on five samples.

Fig. 1a shows that the velocity norm varies substantially along the trajectory. Fig. 1b and Fig. 1c further demonstrate that $v_\theta$ is sensitive to both time perturbations (temporal variation) and state perturbations (spatial variation). These observations indicate nonzero acceleration; hence the Euler update induces integration error that grows with larger $\Delta t$, degrading sample quality. This motivates samplers that explicitly account for acceleration within the integration step.

## 3.2 SECOND-ORDER ACCELERATION-AWARE SAMPLER

The empirical evidence above shows that the learned velocity varies with both time and state, violating the piecewise-constant assumption implicit in Euler updates. To account for the acceleration effects, we augment the sampler with a second-order Taylor expansion of the trajectory around $t$:

$$x_{t+\Delta t} \approx x_t + v_\theta(x_t, t) \cdot \Delta t + \frac{1}{2} \frac{dv_\theta(x_t, t)}{dt} \cdot (\Delta t)^2 \tag{4}$$

where the second-order term $\frac{dv_\theta(x_t, t)}{dt}$ is the total acceleration. By the chain rule, this total acceleration can be further decomposed into temporal and spatial components, reflecting its dependence on both time and state:

$$\frac{dv_\theta(x_t, t)}{dt} = \underbrace{\frac{\partial v_\theta(x_t, t)}{\partial t}}_{\text{temporal acceleration}} + \underbrace{\frac{\partial v_\theta(x_t, t)}{\partial x_t} \cdot \frac{dx_t}{dt}}_{\text{spatial acceleration}} = \frac{\partial v_\theta}{\partial t} + \frac{\partial v_\theta}{\partial x_t} \cdot v_\theta(x_t, t) \tag{5}$$

A detailed derivation is provided in Appendix A. Substituting Eq. 5 into Eq. 4 gives the full second-order update:

$$x_{t+\Delta t} = x_t + v_\theta(x_t, t) \cdot \Delta t + \frac{1}{2} \left( \frac{\partial v_\theta}{\partial t} + \frac{\partial v_\theta}{\partial x_t} v_\theta \right) \cdot (\Delta t)^2 \tag{6}$$

While conceptually straightforward, computing acceleration at every step is nontrivial in high-dimensional image space. The temporal term requires differentiating the network with respect to its time input, and the spatial term is a Jacobian-vector product (JVP). Forming the full Jacobian is intractable for image-sized states. Besides, even efficient JVPs typically require extra differentiation passes, increasing both runtime and memory use per step. Finite differences can be noisy and still incur additional computation. In the following subsections, we develop efficient approximations to both acceleration components to keep the computation close to the first-order sampling while improving integration accuracy.

## 3.3 COMPENSATING TEMPORAL ACCELERATION VIA TIME-SHIFTING

The temporal acceleration term $\frac{\partial v_\theta}{\partial t}$ quantifies the explicit change in the velocity field with respect to time, holding the state fixed. However, directly computing this partial derivative can be costly, as it would require multiple model calls to estimate the finite differences. We propose an efficient approximation inspired by a fundamental observation: the velocity field evolves smoothly with time.

Our key idea is to estimate the change by evaluating the network at a strategically chosen future time point. Specifically, evaluating the velocity at a slightly future time $t + \delta$ for the same state $x_t$ obtains $v_\theta(x_t, t+\delta)$ and the difference between $v_\theta(x_t, t+\delta) - v_\theta(x_t, t)$ approximates the change due purely to time advancement, which is the essence of the temporal acceleration. We can formalize this intuition with a Taylor expansion:

$$v_\theta(x_t, t + \delta) \approx v_\theta(x_t, t) + \frac{\partial v_\theta(x_t, t)}{\partial t} \cdot \delta \tag{7}$$

To integrate this into the second-order update (Eq. 6), we set the shift $\delta = \Delta t/2$. The product of the temporal acceleration and the timestep $\Delta t$ is then approximated by:

$$\frac{\partial v_\theta}{\partial t} \cdot \Delta t \approx 2 \left( v_\theta(x_t, t + \frac{1}{2}\Delta t) - v_\theta(x_t, t) \right) \tag{8}$$

Substituting Eq. 8 into the full second-order update (Eq. 6) simplifies the expression by absorbing the original velocity and the temporal acceleration term into a single, time-shifted velocity evaluation:

$$x_{t+\Delta t} \approx x_t + v_\theta(x_t, t) \cdot \Delta t + \frac{1}{2}\left(\frac{\partial v_\theta}{\partial t} + \frac{\partial v_\theta}{\partial x_t}v_\theta\right) \cdot (\Delta t)^2$$

$$\approx x_t + v_\theta(x_t, t) \cdot \Delta t + \left(v_\theta(x_t, t + \frac{1}{2}\Delta t) - v_\theta(x_t, t)\right) \cdot \Delta t + \frac{1}{2}\frac{\partial v_\theta}{\partial x_t}v_\theta \cdot (\Delta t)^2$$

$$= x_t + v_\theta(x_t, t + \frac{1}{2}\Delta t) \cdot \Delta t + \frac{1}{2}\frac{\partial v_\theta}{\partial x_t}v_\theta \cdot (\Delta t)^2. \tag{9}$$

This result has a compelling interpretation: compensating temporal acceleration is equivalent to using a time-shifted velocity evaluated at the midpoint of the integration interval $t + \Delta t/2$. This is analogous to the midpoint method in numerical ODE integration, which achieves a local truncation error of $\mathcal{O}(\Delta t^3)$ by better capturing the average flow within the step. Our approximation thus reduces the error from the temporal component without explicitly computing expensive temporal derivatives. We further explain the timeshift from the lens of "trajectory switching" in Appendix A.1.

### 3.4 COMPENSATING SPATIAL ACCELERATION VIA SCALE MODULATION

The spatial acceleration term $\frac{\partial v_\theta}{\partial x_t}v_\theta$ is a Jacobian-vector product, which is expensive and numerically unstable to compute exactly in high-dimensional image spaces. We therefore seek an approximation that captures its leading effect while keeping the first-order computational cost.

Fig. 1c indicates that the velocity responds approximately linearly to small perturbations in the state. This suggests preserving the direction of motion while adjusting the step size via speed modulation. To explicitly model the effect brought by the spatial acceleration along the velocity direction, we decompose the spatial acceleration into components parallel and orthogonal to the current velocity direction $v_\theta$:

$$\frac{\partial v_\theta}{\partial x_t}v_\theta = \frac{<\frac{\partial v_\theta}{\partial x_t}v_\theta, v_\theta>}{||v_\theta||_2^2}v_\theta + r_\theta = \frac{v_\theta^\top\left(\frac{\partial v_\theta}{\partial x_t}\right)v_\theta}{||v_\theta||_2^2}v_\theta + r_\theta = \kappa_t v_\theta + r_\theta, \quad r_\theta \perp v_\theta \tag{10}$$

The coefficient $\kappa_t$ is the directional derivative of the velocity along itself, projected back onto the direction of motion. Therefore, retaining only the parallel component corrects how fast we move along the current direction, which directly targets the dominant effect observed in Fig. 1c. The orthogonal component $r_\theta$ is considerably harder to estimate without extra network evaluations or Jacobians. We therefore neglect it for efficiency and stability in the few-step scenario.

Incorporating only the parallel component of the spatial acceleration into Eq. 9 yields:

$$x_{t+\Delta t} \approx x_t + v_\theta(x_t, t + \tfrac{1}{2}\Delta t) \cdot \Delta t + \tfrac{1}{2}\kappa_t v_\theta(x_t, t + \tfrac{1}{2}\Delta t) \cdot (\Delta t)^2, \tag{11}$$

which shows that spatial acceleration acts as a multiplicative correction to the effective step size along the current direction. Defining $\alpha_t \approx 1 + \kappa_t\Delta t/2$, we obtain the practical update:

$$x_{t+\Delta t} = x_t + \alpha_t v_\theta(x_t, t + \tfrac{1}{2}\Delta t) \cdot \Delta t \tag{12}$$

This form preserves the direction given by the time-shifted velocity while compensating for speed changes via a scalar gain. Inspired from the near-linearity in Fig. 1c, we approximate $\alpha_t$ as a smooth, time-dependent schedule that captures the average trend of speed variation along the flow using a simple linear schedule:

$$\alpha_t = \alpha_{\text{start}} + (\alpha_{\text{end}} - \alpha_{\text{start}}) \cdot t, \quad t \in [0, 1]. \tag{13}$$

Based on Eq. 12, we design a cache-based sampler that exploits direction difference to determine the caching strategy. See Appendix A.3 for details.

## 4 EXPERIMENTS

### 4.1 EXPERIMENTAL SETUPS

**Models, Datasets, and Baselines.** We evaluate our method on representative flow-based text-to-image models, namely FLUX-dev (Labs, 2024) and Stable Diffusion 3.5 (Esser et al., 2024). Unlike prior works that typically rely on a single dataset, we conduct a comprehensive evaluation on

Table 1: Quantitative results on FLUX-dev and Stable Diffusion 3.5-large (SD 3.5) across four datasets. The base samplers use 20 and 30 denoising steps respectively. Latency is reported in seconds; subscripts denote speedup relative to each model's base sampler. ▨ indicates metrics where few-step A²S surpasses the corresponding multi-step base. **Bolded** denotes the best score.

| | Method | FLUX-dev (20 steps) | | | | | | SD 3.5-large (30 steps) | | | | | |
|---|---|---|---|---|---|---|---|---|---|---|---|---|---|
| | | Latency(s)↓ | PIQA↑ | AES↑ | CLIP↑ | Pick↑ | HPSv2↑ | Latency(s)↓ | PIQA↑ | AES↑ | CLIP↑ | Pick↑ | HPSv2↑ |
| **MSCOCO** | Base | $15.51_{×1.00}$ | 0.8353 | 6.3547 | 31.4861 | 23.0181 | 0.2894 | $20.52_{×1.00}$ | 0.8222 | 6.2415 | 32.4559 | 22.7901 | 0.2878 |
| | +Δ-DiT | $8.50_{×1.82}$ | 0.7560 | 6.1667 | 31.5278 | 22.7156 | 0.2833 | $7.88_{×2.60}$ | 0.7691 | 6.1425 | 32.2669 | 22.4705 | 0.2824 |
| | +FORA | $8.58_{×1.81}$ | 0.7951 | 6.2795 | 31.5754 | 22.8030 | 0.2849 | $7.55_{×2.72}$ | 0.7612 | 6.1541 | 32.2797 | 22.4651 | 0.2822 |
| | +TeaCache | $6.28_{×2.47}$ | 0.7925 | 6.3327 | 31.5135 | 22.8110 | 0.2841 | $6.84_{×3.00}$ | 0.7946 | 6.1420 | 32.2489 | 22.4778 | 0.2853 |
| | +DPM | $6.26_{×2.48}$ | 0.7866 | 6.2686 | 31.6541 | 22.6635 | 0.2843 | $6.84_{×3.00}$ | 0.7870 | 6.1773 | 32.2498 | 22.5119 | 0.2855 |
| | +UniPC | $6.26_{×2.48}$ | 0.7952 | 6.2663 | 31.6450 | 22.6658 | 0.2857 | $6.89_{×2.98}$ | 0.7921 | 6.1631 | 32.1262 | 22.4750 | **0.2864** |
| | +A²S | $6.19_{×2.51}$ | **0.8132** | **6.4352** | **31.6768** | **22.9733** | **0.2876** | $6.64_{×3.09}$ | 0.7971 | 6.2095 | 32.5638 | 22.6009 | 0.2864 |
| **DiffusionDB** | Base | $15.52_{×1.00}$ | 0.8486 | 7.0069 | 31.9292 | 21.1640 | 0.2741 | $20.63_{×1.00}$ | 0.8152 | 6.7790 | 32.8839 | 21.4617 | 0.2751 |
| | +Δ-DiT | $8.61_{×1.80}$ | 0.7274 | 6.8457 | 31.6846 | 20.6779 | 0.2675 | $7.89_{×2.62}$ | 0.7452 | 6.7195 | 32.8759 | 21.1428 | 0.2684 |
| | +FORA | $8.64_{×1.80}$ | 0.8037 | 6.9249 | 31.9023 | 21.0369 | 0.2700 | $7.54_{×2.73}$ | 0.7390 | 6.7330 | 32.8784 | 21.1306 | 0.2682 |
| | +TeaCache | $6.30_{×2.46}$ | 0.7782 | 7.0054 | 32.1365 | 21.1151 | 0.2688 | $6.94_{×2.97}$ | 0.7278 | 6.7181 | 32.8399 | 21.0646 | 0.2677 |
| | +DPM | $6.28_{×2.47}$ | 0.7829 | 6.9392 | 31.9322 | 20.8954 | 0.2698 | $6.84_{×3.02}$ | 0.7769 | 6.7861 | 33.0042 | 21.2064 | 0.2711 |
| | +UniPC | $6.33_{×2.45}$ | 0.8044 | 6.9492 | 31.8827 | 20.9041 | 0.2713 | $6.89_{×2.99}$ | 0.7892 | 6.7594 | 32.8315 | 21.1673 | 0.2722 |
| | +A²S | $6.23_{×2.49}$ | **0.8054** | **7.1996** | **32.2102** | **21.3643** | **0.2735** | $6.40_{×3.22}$ | 0.7901 | 6.8067 | 33.1371 | 21.2525 | 0.2730 |
| **Pick-a-Pic** | Base | $15.53_{×1.00}$ | 0.8265 | 6.7594 | 31.9251 | 21.2654 | 0.2759 | $20.63_{×1.00}$ | 0.8067 | 6.5171 | 32.5863 | 22.1848 | 0.2757 |
| | +Δ-DiT | $8.59_{×1.81}$ | 0.7524 | 6.5584 | 31.7751 | 21.9596 | 0.2702 | $7.87_{×2.62}$ | 0.7374 | 6.4802 | 32.5863 | 21.9141 | 0.2699 |
| | +FORA | $8.64_{×1.81}$ | 0.7781 | 6.6504 | **31.8977** | 22.0116 | 0.2713 | $7.55_{×2.73}$ | 0.7296 | 6.4898 | 32.5816 | 21.9071 | 0.2697 |
| | +TeaCache | $6.27_{×2.48}$ | 0.7633 | 6.7709 | 31.6654 | 22.0854 | 0.2706 | $6.76_{×3.05}$ | 0.7717 | 6.4776 | 32.5656 | 21.9250 | 0.2715 |
| | +DPM | $6.28_{×2.47}$ | 0.7679 | 6.6501 | 31.7623 | 21.8770 | 0.2708 | $6.84_{×3.02}$ | 0.7626 | 6.4804 | 32.5837 | 21.9336 | 0.2715 |
| | +UniPC | $6.29_{×2.47}$ | 0.7865 | 6.6653 | 31.7649 | 21.8848 | 0.2724 | $6.88_{×3.00}$ | 0.7728 | 6.4504 | 32.5458 | 21.9564 | 0.2725 |
| | +A²S | $6.23_{×2.50}$ | 0.7873 | **6.8760** | 31.7706 | **22.1795** | **0.2735** | $6.35_{×3.25}$ | 0.7864 | 6.5093 | 32.6654 | 21.9928 | 0.2727 |
| **HPDv2** | Base | $15.52_{×1.00}$ | 0.8419 | 6.7367 | 32.5582 | 22.6456 | 0.2836 | $20.66_{×1.00}$ | 0.8089 | 6.5974 | 33.9259 | 22.7529 | 0.2834 |
| | +Δ-DiT | $8.60_{×1.80}$ | 0.7778 | 6.5910 | 32.5681 | 22.4429 | 0.2776 | $7.91_{×2.61}$ | 0.7467 | 6.5005 | 34.0328 | 22.4431 | 0.2777 |
| | +FORA | $8.65_{×1.79}$ | 0.8016 | 6.6672 | 32.6645 | 22.5003 | 0.2790 | $7.57_{×2.73}$ | 0.7448 | 6.5269 | 34.0575 | 22.4321 | 0.2775 |
| | +TeaCache | $6.32_{×2.46}$ | 0.7799 | 6.7492 | 32.7640 | 22.5746 | 0.2782 | $6.96_{×2.97}$ | 0.7302 | 6.5132 | 34.0283 | 22.3817 | 0.2771 |
| | +DPM | $6.29_{×2.47}$ | 0.7869 | 6.6746 | 32.6868 | 22.3607 | 0.2787 | $6.84_{×3.02}$ | 0.7709 | **6.5728** | 34.1335 | 22.5697 | 0.2801 |
| | +UniPC | $6.30_{×2.46}$ | 0.8042 | 6.6806 | 32.7982 | 22.3595 | 0.2801 | $6.86_{×3.01}$ | 0.7766 | 6.5254 | 33.9404 | 22.5020 | 0.2809 |
| | +A²S | $6.21_{×2.50}$ | **0.8052** | **7.0025** | **32.8823** | **22.7729** | **0.2836** | $6.64_{×3.11}$ | 0.7808 | 6.5484 | 34.1450 | 22.5794 | **0.2814** |

four diverse benchmarks: MSCOCO (Lin et al., 2014), DiffusionDB (Wang et al., 2023), Pick-a-Pic (Kirstain et al., 2023), and the Human Preference Dataset v2 (HPDv2) (Wu et al., 2023). These datasets cover different prompt distributions, thereby testing the model's ability to synthesize a broad range of content. For **baselines**, we compare against two major categories: (i) cache-based methods (FORA (Selvaraju et al., 2024), Δ-DiT (Chen et al., 2024), and TeaCache (Liu et al., 2025)); and (ii) fast samplers (DPM (Lu et al., 2022), UniPC (Zhao et al., 2023)). Together, these baselines represent the main directions of few-step approaches. Further details are provided in Appendix B.

**Evaluation Metrics.** We evaluate our method along three key dimensions: (i) **computational efficiency**, measured by per-sample latency (in seconds) and overall speedup (shown in subscript); (ii) **image quality**, assessed using no-reference image quality assessment (IQA) metrics, including PIQA (Zalcher et al., 2025) and AES (LAION-AI, 2022); and (iii) **textual alignment and human preference**, quantified by CLIP (Radford et al., 2021), HPSv2 (Wu et al., 2023), and Pick (Kirstain et al., 2023), which capture the correspondence between generated images, input prompts, and human judgments. More details about these evaluation metrics are provided in Appendix B.2.

**Implementation Details.** All experiments are conducted on a single NVIDIA H20 GPU (96 GB). Our method is implemented by compensating for both temporal and spatial acceleration. For temporal acceleration, we apply a timeshift $\delta$ equal to half the integration step size ($\delta = \frac{1}{2}\Delta t$). Since the base scheduler employs a resolution-dependent rescaling of the timestep schedule (Esser et al., 2024), the integration step size $\Delta t$ varies across steps; we follow the same schedule, making the timeshift $\delta$ step-dependent as well. For spatial acceleration, we use a linear schedule for the scaling factor $\alpha_t$, gradually increasing it from $\alpha_{\text{start}} = 0.95$ to $\alpha_{\text{end}} = 1.05$ throughout the generation process. More details are provided in Appendix B.3.

### 4.2 MAIN RESULTS

**Quantitative Results.** Tab. 1 compares the proposed A²S with the base sampler and other baselines. A²S achieves the highest speedups while preserving, and often improving, image quality. It consistently attains the best or near-best AES across datasets and backbones, frequently surpassing the multi-step base. On FLUX-dev, A²S reduces latency to around 6.2 seconds (about x2.5 speedup over the 20-step base) and simultaneously improves semantic and aesthetic metrics: for MSCOCO

| Base | FORA | TeaCache | DPM | UniPC | Ours |
|------|------|----------|-----|-------|------|

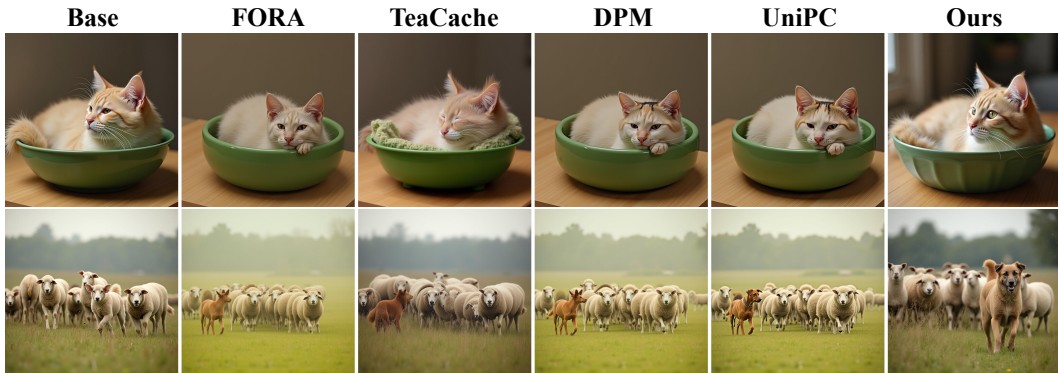

Figure 2: Qualitative comparisons on MSCOCO dataset with FLUX-dev (20 steps) as the base model. Top: *"A cat laying in a green bowl on a wooden table."* Bottom: *"A dog herding a group of sheep in a meadow."* Best viewed when zoomed in for details.

and DiffusionDB, $A^2S$ exhibits higher AES, CLIP and Pick scores than the base. On HPDv2, the AES improves significantly from 6.7367 to 7.0025 despite using far fewer steps. On SD 3.5, the speedup is larger and the image quality is at least on par with the base. On MSCOCO, DiffusionDB and Pick-a-Pic, $A^2S$ reaches the top scores in nearly all metrics. On preference-oriented metrics (Pick, HPSv2), $A^2S$ matches or outperforms all baselines, improving the Pick score on three out of four datasets under FLUX (DiffusionDB, Pick-a-Pic, HPDv2). Overall, $A^2S$ provides the best speed–quality trade-off: it is 2.5–3.2× faster, while often exceeding the multi-step base on alignment and aesthetics and outperforming accelerated samplers on preference metrics.

**Qualitative Results.** We present qualitative comparisons in Fig. 2, where all the baselines and our method employ FLUX-dev as the base model. In the first example, existing methods noticeably reduce visual fidelity and texture richness, resulting in blurred details or unrealistic appearances. Compared to the base model, our method enhances realism by preserving fine-grained fur textures, producing a more informative background, and rendering natural depth-of-field effects that together yield superior visual quality. In the second example, competing methods fail to faithfully preserve the target object, causing the *dog* to vanish or appear unnatural. In contrast, our approach restores the missing semantic element, improves text–image alignment, and introduces richer scene details with a more balanced composition. Additional qualitative results are provided in Appendix C.2.

### 4.3 ABLATION STUDIES

**Ablation on Acceleration Compensation.** Unless otherwise noted, all ablations and analyses in this section are conducted with FLUX-dev on the MSCOCO dataset. Tab. 2 evaluates how the temporal ($Acc_{temp}$) and spatial compensation ($Acc_{spat}$) contribute to quality under an aggressive time budget on MSCOCO. We use FLUX-dev at 20 steps as a reference. Disabling both modules exhibits significant drops in PIQA, Pick and HPSv2

Table 2: Ablation on temporal ($Acc_{temp}$) and spatial ($Acc_{spat}$) acceleration compensation on MSCOCO. The top row reports FLUX-dev at 20 steps. Rows 2-5 show results with various components removed (✗) and active (✔).

| $Acc_{temp}$ | $Acc_{spat}$ | PIQA↑ | AES↑ | Pick↑ | HPSv2↑ |
|:---:|:---:|:---:|:---:|:---:|:---:|
| FLUX-dev (20 steps) | | 0.835 | 6.355 | 23.018 | 0.289 |
| ✗ | ✗ | 0.779 | 6.339 | 22.635 | 0.285 |
| ✔ | ✗ | 0.804 | **6.453** | 22.950 | 0.287 |
| ✗ | ✔ | 0.786 | 6.371 | **23.000** | 0.286 |
| ✔ | ✔ | **0.813** | 6.435 | 22.973 | **0.288** |

relative to 20-step baseline, confirming that naive few-step generation harms both perceptual fidelity and human preference. Adding $Acc_{temp}$ recovers most of the perceptual loss with the AES even surpassing the 20-step baseline (6.453 vs 6.355, the highest AES in the ablation). $Acc_{spat}$ provides complementary gains with the strongest Pick metric among the variants. This suggests $Acc_{spat}$ effectively counters spatial artifacts. Enabling both delivers the best overall trade-off: it attains the highest PIQA and HPSv2 while also reaching a comparable AES and Pick metric. Notably, the full model's AES is marginally below $Acc_{temp}$ only and its Pick metric slightly below $Acc_{spat}$ only, indi-

Table 3: Ablation on the timeshift $\delta$. We compare fixed shifts ($\delta \in [-0.02, 0.04]$), a shift optimized via a genetic algorithm (GA), and the analytic midpoint $\delta = \frac{1}{2}\Delta t$ for compensating temporal acceleration $\text{Acc}_{\text{temp}}$.

| Timeshift $\delta$ | PIQA↑ | AES↑ | Pick↑ | HPSv2↑ |
|---|---|---|---|---|
| -0.02 | 0.5618 | 5.5635 | 21.8433 | 0.2678 |
| 0 | 0.7789 | 6.3385 | 22.6349 | 0.2854 |
| 0.02 | 0.7999 | **6.4799** | 22.7707 | 0.2863 |
| 0.04 | 0.7954 | 6.4269 | 22.0910 | 0.2790 |
| GA | 0.8032 | 6.4231 | 22.6929 | 0.2855 |
| $\frac{1}{2}\Delta t$ | **0.8036** | 6.4526 | **22.9496** | **0.2874** |

Table 4: Ablation on spatial acceleration compensation ($\text{Acc}_{\text{spat}}$) using the first-order linear scale schedule in Eq. 13. We vary $(\alpha_{\text{start}}, \alpha_{\text{end}})$ with all other settings fixed; $\alpha = 1.00$ denotes no compensation.

| $(\alpha_{\text{start}}, \alpha_{\text{end}})$ | PIQA↑ | CLIP↑ | Pick↑ | HPSv2↑ |
|---|---|---|---|---|
| (1.00, 1.00) | 0.8036 | 31.3973 | 22.9496 | 0.2874 |
| (1.05, 1.05) | 0.7422 | 31.3911 | 22.5701 | 0.2850 |
| (0.95, 0.95) | 0.7624 | **31.8186** | 22.9178 | 0.2868 |
| (1.05, 0.95) | 0.7974 | 31.4182 | 22.9116 | 0.2871 |
| (0.95, 1.05) | **0.8132** | 31.6768 | **22.9733** | **0.2875** |

cating that the two compensations target different aspects of perceptual quality (aesthetic fidelity vs. semantic preference) and are best used in concert for balanced performance.

**Ablation on $\text{Acc}_{\text{temp}}$.** We study how the timeshift $\delta$ compensates for temporal acceleration. From Eq. 7, setting $\delta = \frac{1}{2}\Delta t$ allows $v_\theta(x_t, t + \frac{1}{2}\Delta t)$ to implicitly capture the acceleration term in the local Taylor expansion. We compare fixed shifts ($\delta \in \{-0.02, 0, 0.02, 0.04\}$), a shift tuned via a genetic algorithm (GA), and the analytic midpoint $\delta = \frac{1}{2}\Delta t$. As shown in Tab. 3, the timeshift $\delta$ is necessary but direction-sensitive. A negative shift severely degrades all metrics, indicating that evaluating too far "behind" the step leads to instability. A large positive shift ($\delta = 0.04$) hurts the human preference, showing that over-correction is also harmful. The midpoint yields the best overall performance: it achieves the highest PIQA, Pick, and HPSv2 scores, and ranks second on AES. This is consistent with our derivation, which reduces the local truncation error and stabilizes the sampling trajectory. The GA schedule adds tuning complexity without consistent gains. Given its simplicity, robustness, and strong results, we adopt the analytic midpoint for all experiments.

**Ablation on $\text{Acc}_{\text{spat}}$.** We study how the linear scaling schedule $\alpha_t$ in Eq. 13 influences performance. We evaluate five settings: no scaling (1.00→1.00), constant increase (1.05→1.05), constant decrease (0.95→0.95), increase→decrease (1.05→0.95), and decrease→increase (0.95→1.05). As shown in Tab. 4, both constant increase or decrease degrade preference and perceptual metrics. Though constant decrease improves CLIP, it still hurts PIQA and Pick, suggesting better text alignment at the cost of human preference. The decrease→increase schedule achieves the best overall trade-off: it obtains the best PIQA, Pick and HPSv2 with CLIP improved over the baseline (31.6768 vs. 31.3973). This suggests that slowing velocity in early denoising steps stabilizes coarse structures, whereas a later boost helps recover high-frequency details without overshooting.

## 4.4 FURTHER ANALYSIS

**Applying A$^2$S to existing baselines.** A$^2$S is a drop-in module that can be attached to diverse baselines without altering the sampling budget. Tab. 5 reports results on FLUX-dev with MSCOCO for three representative methods, including a cache-based approach (FORA) and two fast samplers (DPM and UniPC), before and after adding A$^2$S. Across all methods and metrics, A$^2$S yields consistent improvements. The largest absolute gains are observed on perceptual and aesthetic metrics: for example, with FORA, PIQA rises from 0.795 to 0.844 and AES from 6.280

Table 5: Plug-and-play gains from A$^2$S across samplers. Adding A$^2$S to FORA (N=2), DPM (10-steps), and UniPC (10-steps) consistently improves image quality under identical time budgets.

| Method | PIQA↑ | AES↑ | Pick↑ | HPSv2↑ |
|---|---|---|---|---|
| FORA (N=2) | 0.795 | 6.280 | 22.803 | 0.285 |
| +A$^2$S | 0.844 | 6.459 | 22.876 | 0.288 |
| DPM (10-steps) | 0.810 | 6.301 | 22.827 | 0.287 |
| +A$^2$S | 0.833 | 6.404 | 22.949 | 0.289 |
| UniPC (10-steps) | 0.813 | 6.294 | 22.839 | 0.288 |
| +A$^2$S | 0.838 | 6.460 | 22.849 | 0.290 |

to 6.459. Preference-oriented metrics also benefit: with DPM, Pick increases from 22.827 to 22.949. These results demonstrate that A$^2$S corrects acceleration-induced errors independent of the solver's design and complements a wide range of inference schemes, enhancing visual fidelity, aesthetics, and preference alignment within the same step budgets. The breadth and consistency of these gains

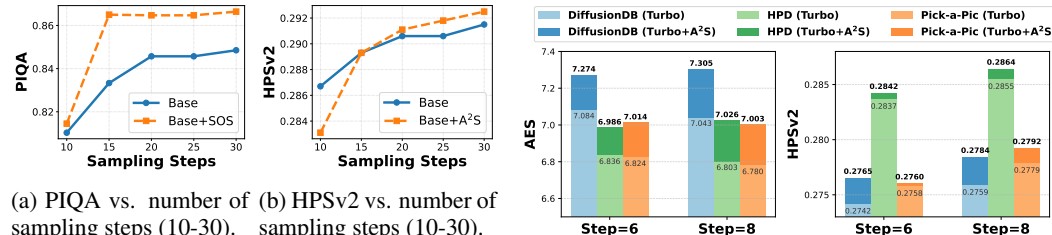

(a) PIQA vs. number of sampling steps (10-30).  (b) HPSv2 vs. number of sampling steps (10-30).

Figure 3: Performance across sampling steps.  Figure 4: Performance on Distilled FLUX-Turbo-$\alpha$.

establish A$^2$S as a general-purpose, plug-and-play enhancement readily integrable into existing text-to-image generation pipelines.

**Performance across timestep budget.** We evaluate our method on FLUX-dev under varying sampling steps (10–30 steps) on MSCOCO to analyze behavior in both few-step and multi-step settings. As shown in Fig. 3, A$^2$S consistently improves PIQA at all step counts. We also note that after applying A$^2$S, PIQA at 15 steps already surpasses the base at 30 steps, indicating roughly 2x efficiency for PIQA. For HPScore, the effect is step-dependent: A$^2$S is slightly lower at 10 steps, almost equal at 15 steps, and yields small but consistent improvements from 20 to 30 steps. The base curves show diminishing gains beyond 20 steps (PIQA plateau and gently rising HPSv2). A$^2$S lifts the curves upward and keeps PIQA improving with steps, indicating that correcting acceleration lets additional steps translate into real quality gains instead of saturating early. These results indicate that while A$^2$S is tailored for few-step sampling, it transfers well to multi-step context: it provides immediate PIQA gains even at very low budgets and improves HPSv2 once the step budget is moderate.

**Applying A$^2$S to distilled few-step models.** We evaluate A$^2$S on a distilled few-step model (FLUX-Turbo-$\alpha$ (Alimama-Creative Team, 2024)) with 6 and 8 denoising steps across DiffusionDB, HPDv2, and Pick-a-Pic, as shown in Fig. 4. The figure contains two grouped bar plots (AES and HPSv2), each with two step budgets (6 and 8). Within each group, the light bar shows the original turbo baseline, and the darker stacked segment shows the incremental gain from our method. This visualization makes both the absolute performance and the plug-in gain clear at fixed compute. It can be observed that A$^2$S consistently improves both aesthetics (AES) and human preference (HPSv2). At 6 steps, AES increases by +0.190 (DiffusionDB), +0.150 (HPDv2), and +0.189 (Pick-a-Pic); at 8 steps, the gains grow to +0.263 (DiffusionDB), +0.223 (HPDv2), and +0.223 (Pick-a-Pic), respectively. HPSv2 shows smaller but uniform improvements. The larger gains at 8 steps suggest that A$^2$S is especially effective when the sampler has a slightly longer refinement horizon, allowing temporal midpoint correction and progressive spatial scaling to take fuller effect. Notably, the same A$^2$S configuration is used across datasets and step budgets, yet yields consistent benefits, indicating strong plug-and-play generalization across data distributions, prompts, and metrics.

## 5 CONCLUSION

In summary, this work addresses the challenge of preserving high image quality in few-step generation for rectified flow models, a limitation of first-order samplers that assume zero acceleration. By identifying degradation as a consequence of neglecting second-order dynamics in the velocity field, we propose **A$^2$S**, a lightweight, training-free sampler that explicitly models temporal and spatial acceleration. A$^2$S introduces two corrections: a time-shifted velocity evaluation for temporal acceleration and a scalar modulation for spatial acceleration. These modifications require no extra model calls, making A$^2$S a plug-and-play enhancement for existing RF models and sampling schedules. Empirically, A$^2$S demonstrates substantial improvements in image quality and stability across multiple pretrained RF models and benchmarks, particularly in the few-step setting.

Our contributions advance the understanding of RF sampling dynamics and provide a practical solution to bridge the efficiency-accuracy gap in real-world applications. Moreover, A$^2$S opens avenues for exploring higher-order effects in continuous-time flows and their implications. By decoupling acceleration components and demonstrating their benefits with minimal overhead, A$^2$S provides a foundation for robust, efficient sampling strategies in resource-constrained settings.

**Reproducibility Statement.** We have made extensive efforts to ensure the reproducibility of our work. All models, datasets, and baseline methods used in our experiments are described in Sec. 4.1. Further implementation details, including experiment design and hyperparameters of $A^2S$, are provided in Appendix B. Source code containing our method implementation and evaluation scripts will be released soon. These resources collectively facilitate the reproduction of all results reported in this paper.

**Ethical Statement.** This work does not involve human subjects, personal or sensitive data, or practices that raise concerns under the ICLR Code of Ethics. We identify no potential harms, conflicts of interest, discrimination or fairness issues, privacy or security risks, legal implications, or threats to research integrity arising from this study. Accordingly, we affirm that this research complies fully with the ICLR Code of Ethics.

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

## A  FURTHER EXPLANATION ON ACCELERATION DECOMPOSITION

We further clarify Eq. 5 in the main text. The total acceleration can be divided into these two components because the learned velocity $v_\theta(x_t, t)$ depends both on state $x_t$ and the time $t$, while $x_t$ itself is a function of t. This dependence is illustrated in Fig. 5. For intuition, consider the simple bilinear form:

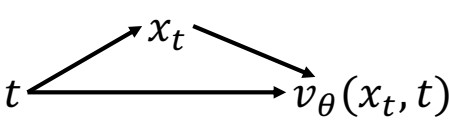

Figure 5: Dependency between $t$, $x_t$, and $v_\theta$.

$$v_\theta(x_t, t) = Ax_t + Bt + Cx_t \cdot t \tag{14}$$

where $A$, $B$, and $C$ are constants (or tensors) with compatible dimensions. Direct differentiation with respect to $t$ gives:

$$\frac{dv_\theta(x_t, t)}{dt} = A\frac{dx_t}{dt} + B + C(\frac{dx_t}{dt}t + x_t) \tag{15}$$

$$= (A + Ct)\frac{dx_t}{dt} + Cx_t + B \tag{16}$$

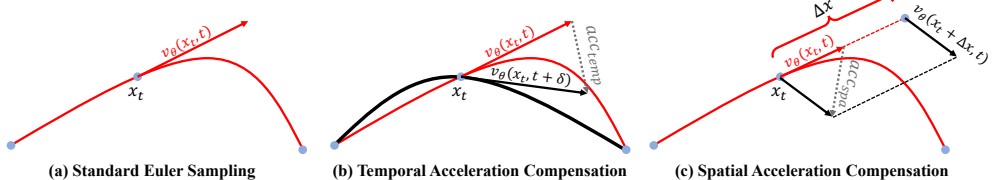

Figure 6: A intuitive illustration on temporal and spatial acceleration.

whereas computing the partial derivatives yields:

$$\frac{\partial v_\theta(x_t, t)}{\partial t} = B + Cx_t \quad \frac{\partial v_\theta(x_t, t)}{\partial x_t} = A + Ct \tag{17}$$

Substituting Eq. 17 into Eq. 5 gives:

$$\frac{dv_\theta(x_t, t)}{dt} = \frac{\partial v_\theta(x_t, t)}{\partial t} + \frac{\partial v_\theta(x_t, t)}{\partial x_t} \cdot \frac{dx_t}{dt} \tag{18}$$

$$= (B + Cx_t) + (A + Ct)\frac{dx_t}{dt} \tag{19}$$

which mathes the result obtained by direct differentiation (Eq. 16).

## A.1 TEMPORAL ACCELERATION

We further elaborate on temporal acceleration. As defined in Eq. 5 and illustrated in Fig. 6(b), the temporal acceleration can be interpreted as the difference between $v_\theta(x_t, t)$ and $v_\theta(x_t, t + \delta)$: it measures how the predicted velocity changes under a small perturbation $\delta$ in time while holding the state $x_t$ fixed. To build intuition, we adopt a trajectory-switching perspective. In Fig. 6(b), the red curve denotes the original Euler sampling trajectory, which reaches the current state $x_t$ at time $t$. Consider instead the black trajectory, which reaches the same state $x_t$ at time $t + \delta$. For the black trajectory, the current time is $t + \delta$, and we have:

$$x_t^{red} = x_t = x_{t+\delta}^{black} \tag{20}$$

Hence, the predicted velocity at time $t + \delta$ along the black trajectory satisfies:

$$v_\theta(x_{t+\delta}^{black}, t + \delta) = v_\theta(x_t, t + \delta) \tag{21}$$

This shows that replacing $v_\theta(x_t, t)$ by $v_\theta(x_t, t + \delta)$ implicitly switches the sampler from the red trajectory to the black one. From this viewpoint, temporal acceleration compensation can be understood as shifting the sampling trajectory forward in time, potentially altering the direction of the predicted velocity.

## A.2 SPATIAL ACCELERATION

We next detail the spatial acceleration. As defined in Eq. 5 and illustrated in Fig. 6(c), the spatial acceleration quantifies how the predicted velocity responds to changes in the state **along the direction of the current velocity**, with time held fixed. For a small step size $\tau > 0$, take a forward step along the current velocity to obtain the state perturbation: $\Delta x = \tau \cdot v_\theta$. The spatial acceleration can then be viewed as the difference: $v_\theta(x_t + \Delta x, t) - v_\theta(x_t, t)$, which, to first order, equals $\tau \frac{\partial v_\theta(x_t, t)}{\partial x_t} v_\theta$ i.e., the Jacobian–vector product term in Eq. 5.

## A.3 CACHE-BASED METHOD

Equipped with the proposed A²S, We design a cache-based sampler that exploits direction difference to determine the caching strategy (Alg. 1). Specifically, inspired by TeaCacheLiu et al. (2025), we leverage the cosine similarity between the input and output to define dynamic caching policies.

---

**Algorithm 1** Cache-Based A$^2$S Sampling for Rectified Flow

---

**Require:** Rectified Flow Model $v_\theta$; N-Step Timestep Schedule $\{t_i\}_{i=0}^N$; threshold $\theta$

 1: $k \leftarrow 0; h \leftarrow 1$          ▷ Initialize the step index $k$ and the step size $h$
 2: $v_{prev} \leftarrow$ null            ▷ Initialize the cached velocity $v_{prev}$
 3: $x_0 \sim \mathcal{N}(0,1)$         ▷ Initialize the image from Gaussian Distribution
 4: $\Delta t = t_0/2$            ▷ Initialize the timeshift, Sec. 3.3
 5: **while** $k < N$ **do**
 6:   $v_k \leftarrow (1 + \alpha_{t_k}) \cdot v_\theta(x_{t_k}, t_k + \frac{1}{2}\Delta t)$    ▷ A$^2$S Sampling (Eq. 12 in the main text)
 7:   **if** $\text{CosSim}(v_{prev}, v_k) > \theta$ **then**   ▷ Compare current velocity to the cached one
 8:     $h \leftarrow h + 1$         ▷ Adaptive step-size update
 9:   $k_{next} \leftarrow \min(k + h, N)$       ▷ Jump to the next index
10:   $\Delta t \leftarrow t_{k_{next}} - t_k$        ▷ Update the time interval
11:   $x_{k_{next}} \leftarrow x_k + v_k \cdot \Delta t$      ▷ Rectified flow update
12:   $v_{prev} \leftarrow v_k; \quad k \leftarrow k_{next}$   ▷ Update the cached velocity and the index
13: **return** $x_{k_{next}}$

---

# B MORE DETAILS ON EXPERIMENTAL SETUPS

## B.1 DATASETS

We evaluate our method on four different datasets, including MSCOCO (Lin et al., 2014), DiffusionDB (Wang et al., 2023), Pick-a-Pic (Kirstain et al., 2023), and the HPDv2 (Wu et al., 2023).

**MSCOCO.** MSCOCO (Lin et al., 2014) is a large-scale dataset of everyday scenes containing common objects in natural context. Collected from the web with an emphasis on non-iconic views, it features multiple objects interacting within each scene. Each data sample is a real image paired with rich human annotations: object bounding boxes and instance segmentation masks for 80 "thing" categories (with additional "stuff" and panoptic labels in later releases), five human-written captions per image, and person keypoints for the people subset. Spanning hundreds of thousands of images and well over a million annotated object instances, MSCOCO provides standardized train/val/test splits and serves as a cornerstone benchmark for multiple vision tasks. Its advantage lies in the diversity and contextual complexity of scenes, which encourages models to reason about objects in realistic settings rather than isolated, iconic views. In this paper, we utilize its validation set for our evaluation.

**DiffusionDB.** DiffusionDB (Wang et al., 2023) is an open, large-scale gallery of text-to-image generations created by real users of latent diffusion models (notably Stable Diffusion). The dataset is gathered from public web interfaces and community galleries where users share generations. Each data sample includes a user-written text prompt, the synthesized image, and, for many entries, generation metadata such as sampler type, guidance scale, number of steps, seed, and resolution. DiffusionDB contains millions of prompt–image pairs (approximately 2M in the initial release, with subsequent expansions exceeding 10M), covering a wide range of subjects, artistic styles, and parameter settings. Its advantage is that it captures real-world prompting behavior and model usage at scale, enabling research on prompt engineering, controllable generation, and evaluation grounded in authentic user data rather than curated lab settings. In this paper, we randomly select 5000 prompts to evaluate our method.

**Pick-a-Pic.** Pick-a-Pic (Kirstain et al., 2023) is an open, crowdsourced dataset of human preferences for text-to-image generation. Collected through an interactive web interface where users compare multiple images produced for the same prompt, the dataset provides pairwise (and sometimes "both/neither") judgments that indicate which image better matches the user's preference and the prompt. Each entry includes a natural-language prompt, two generated images, a user choice label, and, when available, generation metadata (e.g., model, sampler, guidance scale, steps, and seed). The data spans a broad distribution of subjects, styles, and models (e.g., Stable Diffusion variants and other contemporary systems), yielding hundreds of thousands of comparisons across a diverse prompt set. Pick-a-Pic has become a standard resource for learning and evaluating preference-aligned reward models and metrics, as it reflects real user judgments rather than proxy heuristics. In this paper, we use the official release and randomly select 5000 prompts for evaluation.

**Human Preference Dataset v2 (HPD v2).** HPDv2 (Wu et al., 2023) is a large-scale, curated corpus of human preference annotations for text-to-image generations, released alongside the Human Preference Score v2 (HPSv2) metric. The dataset aggregates pairwise comparisons over images generated from diverse prompts and a variety of models, with labels reflecting users' holistic judgments of prompt adherence, visual quality, and overall appeal. Each sample comprises a prompt, two candidate images, and a binary preference label (optionally including ties/abstentions in some subsets), with standardized formatting to facilitate training preference models and evaluating alignment. HPDv2 covers a wide range of concepts and visual styles and contains on the order of hundreds of thousands to nearly a million comparisons, making it well-suited for learning robust, human-aligned reward functions and metrics. In our experiments, we report results on the HPDv2 test splits.

### B.2 EVALUATION METRICS

Five evaluation metrics are adopted to assess the effectiveness of our proposed method, namely PIQA (Zalcher et al., 2025), AES (LAION-AI, 2022), CLIP (Radford et al., 2021), HPSv2 (Wu et al., 2023) and Pick (Kirstain et al., 2023).

**AES.** AES (Aesthetic Score) (LAION-AI, 2022) estimates the visual appeal of an image using a model trained on datasets with human-provided aesthetic ratings, built on top of CLIP-like image embeddings. It outputs a single scalar score that reflects perceived attractiveness, composition, and overall visual quality and serves as a proxy for the aesthetic quality of generated images and reflects the human aesthetic preferences.

**CLIP.** CLIP (Radford et al., 2021) measures text–image semantic alignment by computing the cosine similarity between CLIP text and image embeddings, typically with a normalization and scaling factor. Originally proposed as a reference-free metric for image captioning, it is widely used in text-to-image evaluation as a proxy for prompt adherence. It outputs a single scalar score that captures semantic consistency.

**HPSv2.** HPSv2 (Wu et al., 2023) is a learned metric designed to approximate human judgments of text-to-image outputs. Trained on broad, curated human preference data spanning diverse prompts and model families, it uses a vision–language backbone with a lightweight prediction head to output a scalar score for each prompt–image pair. The score reflects both prompt adherence and perceived visual quality, and HPSv2 has demonstrated strong cross-model validity and robustness compared to earlier automatic metrics.

**Pick.** Pick is the abbreviation for PickScore (Kirstain et al., 2023). PickScore estimates human preference for a text–image pair using a model trained on large-scale, pairwise human comparisons (e.g., from the Pick-a-Pic dataset). Built on top of CLIP-like image–text embeddings, it predicts which image a human would prefer for a given prompt and outputs a single scalar that integrates factors such as semantic alignment, aesthetic appeal, and overall perceptual quality. It serves as a reference-free proxy for human judgments and has been shown to correlate well with crowd preferences across diverse generators.

**PIQA.** PIQA (Zalcher et al., 2025) is a no-reference image quality assessment (IQA) metric that leverages the CLIP vision transformer as a strong perceptual prior for predicting human judgments of image quality. It employs a lightweight adaptation strategy, using Low-Rank Adaptation (LoRA) to fine-tune only the attention layers of the CLIP visual encoder. This design preserves the rich perceptual knowledge embedded in CLIP—gained from its training on human-written captions that include subjective sentiments and preferences—while allowing for efficient task-specific adaptation. The model outputs a continuous scalar score that aligns with human mean opinion scores (MOS), reflecting perceived quality degradations from both authentic (e.g., motion blur, overexposure) and synthetic (e.g., compression artifacts, noise) distortions.

### B.3 BASELINES AND IMPLEMENTATION DETAILS

All experiments are conducted at the resolution of 1024×1024 using the PyTorch framework with the `diffusers` library from Huggingface for model loading and inference. We adopt the official implementation of SD 3.5-large and FLUX-dev for all evaluations. For conditional generation, we adopt classifier-free guidance (CFG) with a consistent guidance scale of 3.5 across all models. For

Table 6: Quantitative results for both the baselines and our method across a range of hyperparameters on MSCOCO dataset.

| Method | Latency(s)↓ | Speedup↑ | PIQA↑ | AES↑ | CLIP↑ | Pick↑ | HPSv2↑ |
|---|---|---|---|---|---|---|---|
| **FLUX-dev (20 steps)** | 15.51 | x1.00 | 6.3547 | 6.3547 | 31.4861 | 23.0181 | 0.2894 |
| **FORA ($\mathcal{N} = 2$)** | 8.58 | x1.81 | 0.7951 | 6.2795 | 31.5754 | 22.8030 | 0.2849 |
| **FORA ($\mathcal{N} = 3$)** | 6.24 | x2.48 | 0.7344 | 6.1980 | 31.5220 | 22.5226 | 0.2798 |
| **DPM (10 steps)** | 7.76 | x2.00 | 0.8102 | 6.3005 | 31.6443 | 22.8270 | 0.2867 |
| **DPM (8 steps)** | 6.26 | x2.48 | 0.7866 | 6.2686 | 31.6541 | 22.6635 | 0.2843 |
| **UniPC (10 steps)** | 7.77 | x2.00 | 0.8126 | 6.2944 | 31.6343 | 22.8391 | 0.2877 |
| **UniPC (8 steps)** | 6.26 | x2.48 | 0.7952 | 6.2663 | 31.6450 | 22.6658 | 0.2857 |
| **TeaCache ($\rho = 0.45$)** | 7.67 | x2.02 | 0.8077 | 6.4842 | 31.5260 | 22.9164 | 0.2860 |
| **TeaCache ($\rho = 0.50$)** | 7.10 | x2.18 | 0.7978 | 6.4643 | 31.5472 | 22.8980 | 0.2855 |
| **TeaCache ($\rho = 0.55$)** | 6.84 | x2.27 | 0.7946 | 6.4537 | 31.5489 | 22.8778 | 0.2853 |
| **TeaCache ($\rho = 0.60$)** | 6.28 | x2.47 | 0.7925 | 6.4352 | 31.5135 | 22.8110 | 0.2841 |
| **A$^2$S ($\theta = 0.98$)** | 7.64 | x2.03 | 0.8243 | 6.4842 | 31.6774 | 23.0936 | 0.2883 |
| **A$^2$S ($\theta = 0.97$)** | 6.78 | x2.29 | 0.8189 | 6.4643 | 31.6825 | 22.9827 | 0.2882 |
| **A$^2$S ($\theta = 0.96$)** | 6.48 | x2.39 | 0.8163 | 6.4537 | 31.6638 | 22.9777 | 0.2879 |
| **A$^2$S ($\theta = 0.95$)** | 6.19 | x2.51 | 0.8132 | 6.4352 | 31.6774 | 22.9733 | 0.2876 |

SD 3.5, CFG requires two model evaluations, whereas for FLUX-dev it requires only one because CFG is distilled during training.

We evaluate two categories of baselines: (i) cache-based method, including FORA (Selvaraju et al., 2024), $\Delta$-DiT (Chen et al., 2024), TeaCache (Liu et al., 2025). (ii) fast samplers, including DPM (Lu et al., 2022) and UniPC (Zhao et al., 2023). Below we summarize each method and the hyperparameters used in our experiments.

**FORA.** FORA (Selvaraju et al., 2024) caches and reuses the residual outputs of attention layers across timesteps. A single hyperparameter $\mathcal{N}$ controls the cache refresh interval (how frequently the cache is recomputed.) *We set $\mathcal{N} = 2$ for FLUX-dev and $\mathcal{N} = 3$ for SD 3.5.*

**$\Delta$-DiT.** Similar to FORA, $\Delta$-DiT (Chen et al., 2024) leverages residuals, but it further accounts for layer-wise functional difference. It splits the time horizon into two phases and applies distinct caching policies in each phase. The main hyperparameters are the fraction of cached layers, the cache interval $\mathcal{N}$ and the split time. *We cache $\frac{2}{3}$ of the layers, set the split time to the midpoint (10 for the 20-step FLUX-dev schedule and 15 for the 30-step SD 3.5 schedule), and use $\mathcal{N} = 3$ for FLUX-dev and $\mathcal{N} = 4$ for SD 3.5.*

**TeaCache.** TeaCache (Liu et al., 2025) adapts the caching decision based on input-output similarity, measured by the L2 distance. A threshold $\rho$ determines whether to reuse the cache or recompute. *We set $\rho = 0.60$ for FLUX-dev and $\rho = 0.55$ for SD 3.5.*

**DPM and UniPC.** DPM (Lu et al., 2022) and UniPC (Zhao et al., 2023) are both higher order integrator methods for ODEs designed to improve the accuracy and stability under aggressive step sizes. UniPC further adopts a unified predictor-corrector scheme that enhances stability at very low numbers of function evaluation. *We use 8 steps for FLUX-dev and 10 steps for SD 3.5.*

Our method follows a TeaCache-style adaptive caching strategy augmented with the proposed A$^2$S. It requires a threshold $\theta$ (Alg. 1); *we set $\theta = 0.95$.*

## C ADDITIONAL EXPERIMENTAL RESULTS

### C.1 ADDITIONAL QUANTITATIVE RESULTS

To further assess the effectiveness of our approach, we report comprehensive quantitative results for both the baselines and our method across a range of hyperparameters. Tab. 6 summarizes the speed–quality trade-off, further validating the advantages of our method.

Table 7: Performance with and without $A^2S$ in extreme-few step setting.

| Method | PIQA↑ | AES↑ | Pick↑ | HPSv2↑ |
|---|---|---|---|---|
| FLUX-dev (5 steps) | 0.6410 | 6.0621 | 22.0152 | 0.2744 |
| FLUX-dev (5 steps)+$A^2S$ | 0.4819 | 4.9732 | 18.7005 | 0.2330 |
| FLUX-turbo (4 steps) | 0.7722 | 6.3698 | 22.8929 | 0.2846 |
| FLUX-turbo (4 steps)+$A^2S$ | 0.5783 | 6.1073 | 20.9756 | 0.2674 |

## C.2 ADDITIONAL QUALITATIVE RESULTS

We visualize more generated images and provide a comprehensive qualitative comparison across multiple baselines and our method (Fig. 7 and Fig. 8).

## D LIMITATION AND FUTURE WORKS

In this work, we examine second-order effects—specifically, acceleration—in the sampling dynamics of rectified-flow models. We decompose the acceleration into temporal and spatial components and approximate both with lightweight estimators. We identify two main limitations and outline corresponding directions for improvement. First, our method omits the orthogonal component of the spatial acceleration term ($r_\theta$ in Eq. 10). Incorporating this component is expected to improve sampling accuracy; future work will focus on efficient estimation strategies that capture $r_\theta$ without incurring substantial computational overhead. Second, while $A^2S$ is effective overall, we observe severe performance degradation when the number of sampling steps is reduced to about five. As shown in Tab. 7, the degradation becomes pronounced at four steps. Addressing this extreme few-step setting is an important direction for future work, including the exploration of adaptive step-size schedules, higher-order integration schemes, and training-time regularization tailored to low-step sampling.

## E USE OF LARGE LANGUAGE MODELS

In compliance with the conference policy on the use of large language models (LLMs), we disclose that LLMs were used exclusively for writing-related purposes. Specifically, LLMs assisted in:

- Improving grammar and fluency;
- Polishing sentence structure and readability;
- Suggesting alternative phrasings for clarity and conciseness.

No part of the research design, theoretical development, experiments, data analysis, or result interpretation relied on LLMs. All scientific contributions, ideas, and conclusions were conceived and validated solely by the authors.

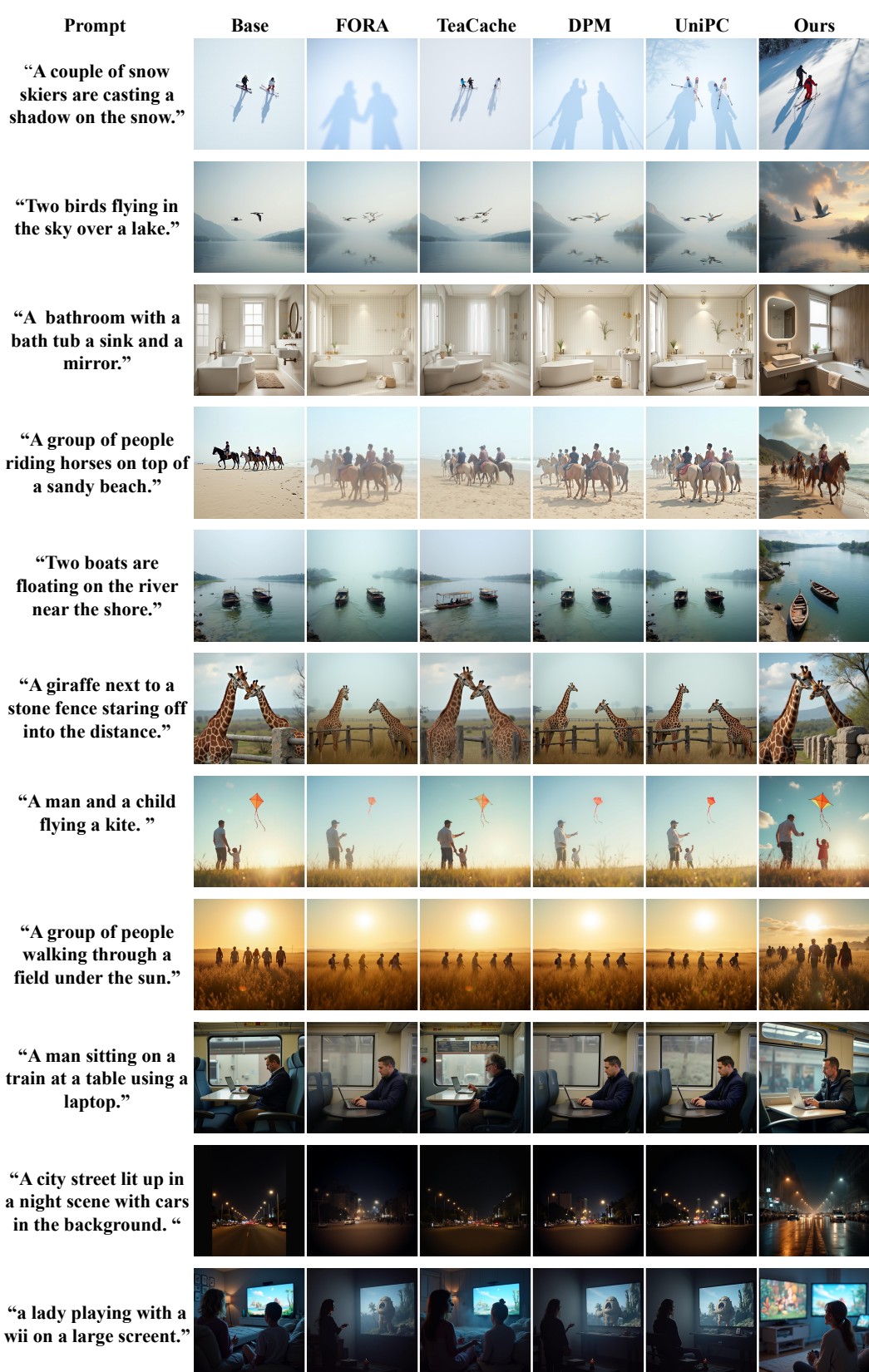

Figure 7: Additional qualitative comparisons with FLUX-dev as the base model.

972
973
974
975
976
977
978
979
980
981
982
983
984
985
986
987
988
989
990
991
992
993
994
995
996
997
998
999
1000
1001
1002
1003
1004
1005
1006
1007
1008
1009
1010
1011
1012
1013
1014
1015
1016
1017
1018
1019
1020
1021
1022
1023
1024
1025

| Prompt | Base | FORA | TeaCache | DPM | UniPC | Ours |
|---|---|---|---|---|---|---|

**"Animation still frame of an attractive female sorceress ( long dark hair ) casting a icy frost spell."**

**"A drawing of a fossil from the burgess shale."**

**"creepy Christmas, trending on unsplash, professional photography, overhead view of a table."**

**"A painting of a smiling woman wearing a shawl, an ultrafine detailed painting by marguerite zorach."**

**"A witch emanating magic from her palms, illuminating the area, closeup."**

**"A wind spirit by ross tran, ethereal, highly detailed, oil on canvas."**

**"Andromeda galaxy in a sphere, transparent clear see - through image from the james webb telescope."**

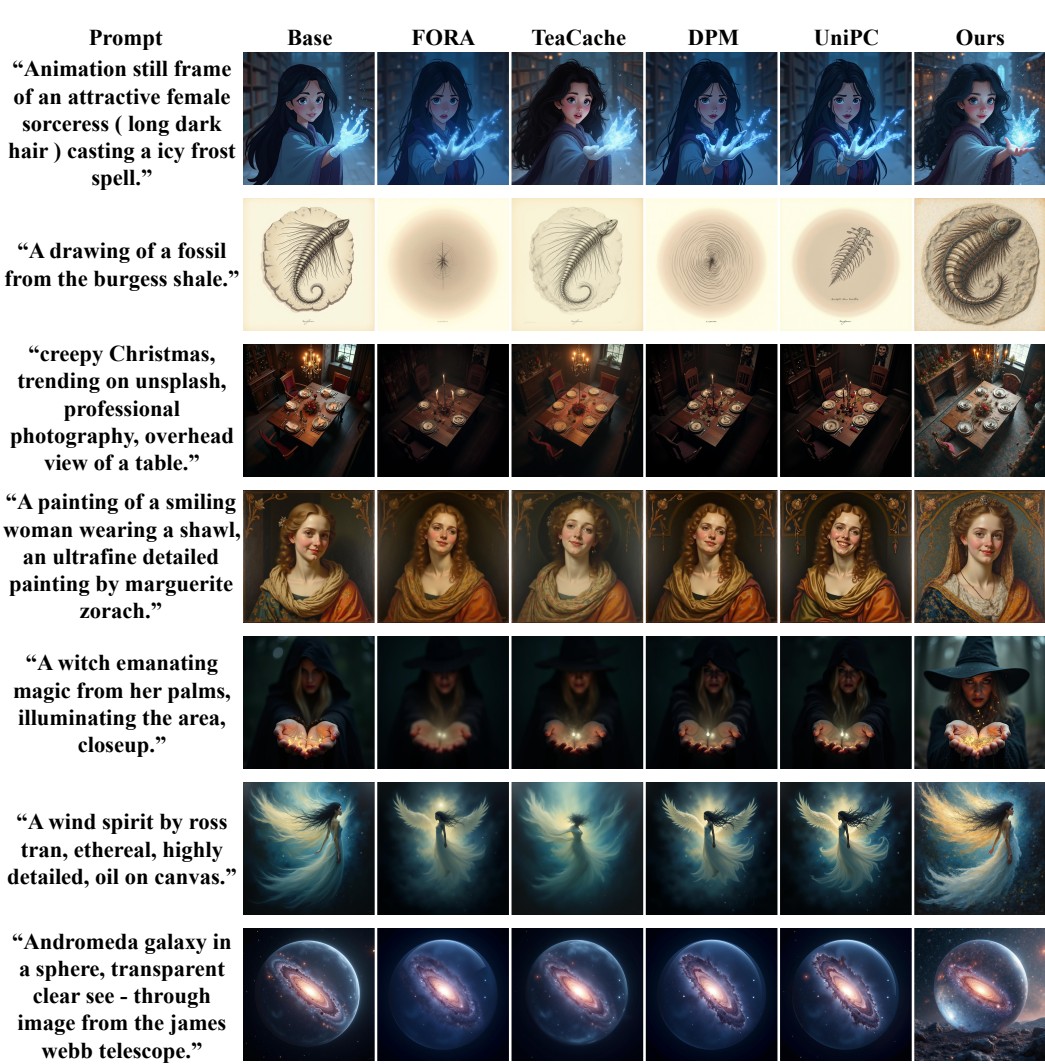

Figure 8: Additional qualitative comparisons with FLUX-dev as the base model.

