# OpenReview forum: "Acceleration-Aware Sampling for Few-Step Rectified Flow Models"
_ICLR.cc/2026/Conference — ICLR 2026 Conference Withdrawn Submission_

### Official Review · Reviewer_xFyh · 2025-10-26

**Soundness:** 2
**Presentation:** 2
**Contribution:** 2
**Rating:** 2
**Confidence:** 4

**Summary:**

The commonly used Euler sampler ignores acceleration, leading to significant discretization errors that dominate few-step sampling. This paper introduces Acceleration-Aware Sampling (AAS), which explicitly models acceleration without increasing computational cost. From a second-order perspective, AAS decomposes acceleration into temporal and spatial components, enabling time-shifted velocity evaluation and adaptive step-size modulation. The method can be seamlessly applied to pretrained rectified-flow models without any retraining.

**Strengths:**

- The paper analyzes few-step degradation in rectified flow models and attributes it to the zero-acceleration assumption of first-order samplers.
- It proposes an Acceleration-Aware Sampler (AAS) that accounts for temporal and spatial acceleration through a time-shift correction and lightweight scale modulation, without requiring retraining. This plug-and-play property enhances practical usability, particularly for large pretrained models where retraining is infeasible (e.g., FLUX-dev).
- The method shows moderate quality improvements in few-step sampling across pretrained rectified-flow models, with minimal implementation overhead.

**Weaknesses:**

- In Figure 1, the evidence is based on only five samples and qualitative trends. Without statistical aggregation or variance analysis, the claim of “non-zero acceleration” lacks quantitative support. Moreover, the results do not show how this non-constancy leads to actual Euler integration error or degradation in generated image quality.
- The proposed method may require re-tuning step-size schedules or scaling coefficients to maintain stability across different pretrained models, which partially limits its claimed zero-overhead generality.
- The ablation study in Section 4.3 does not clearly demonstrate the advantages of modeling temporal and spatial acceleration. The observed trade-offs and gains are inconsistent and lack a clear trend, suggesting that the improvements may stem from empirical hyperparameter tuning rather than a principled effect of the proposed method. Moreover, Table 1 does not provide strong evidence that the method consistently outperforms the baseline.

**Questions:**

- It would strengthen Figure 1 if the authors could quantify an acceleration proxy to provide more concrete evidence of non-zero acceleration.
- In Figure 3(a), please check whether the legend label “Base + SOS” is a typo or intentional, as it is unclear from the context.

---

### Official Review · Reviewer_T23C · 2025-10-27

**Soundness:** 2
**Presentation:** 2
**Contribution:** 2
**Rating:** 4
**Confidence:** 4

**Summary:**

The paper identifies that rectified flow (RF) samplers, when run with very few integration steps, suffer from degraded image quality because the standard Euler integration assumes zero acceleration. The authors derive a second‑order Taylor expansion of the RF trajectory that decomposes acceleration into temporal and spatial components and propose a lightweight acceleration‑aware sampler (A2S) that compensates for both without additional model evaluations. The idea of using a time‑shifted velocity evaluation at mid‑step to approximate temporal acceleration and a simple gain schedule to approximate spatial acceleration is creative in the context of RF sampling, although reminiscent of classical midpoint/RK2 integrators.

**Strengths:**

- Quality: The paper provides a clear derivation of the second‑order update and shows how the acceleration can be decomposed via the chain rule. It proposes practical approximations for the temporal term (midpoint evaluation) and the spatial term (scalar gain) with no extra model calls. Experiments evaluate the method on two text‑to‑image models (FLUX‑dev and Stable Diffusion 3.5) and four benchmarks (MSCOCO, DiffusionDB, Pick‑a‑Pic, HPDv2), comparing against baseline samplers such as Δ‑DiT, FORA, TeaCache, DPM, UniPC and the base Euler sampler. Quantitative results show that A2S achieves ~2.5–3.2× speedups while often improving or matching image quality metrics and preference scores. Ablations isolate the contributions of the temporal and spatial corrections, demonstrating that each component adds measurable improvements. Additional ablations examine the effect of different time shifts and scaling schedules.

- Clarity: The high‑level motivation and derivations are articulated clearly. Key equations are presented step‑by‑step and the proposed update rule is easy to implement.

- Significance: In practical deployment of generative models, latency budgets often require sampling with very few steps, and even small quality improvements are valuable.

**Weaknesses:**

- Limited novelty: The core idea—using a midpoint evaluation to approximate second‑order dynamics—is essentially the classical midpoint method in numerical ODEs. Similarly, modulating the step size based on a linear schedule is a heuristic; the paper does not investigate more principled estimates of the spatial Jacobian. Prior works on efficient diffusion sampling (DPM‑Solver++, DEIS, UniPC, etc.) already employ higher‑order solvers and learned timesteps, and the authors’ method can be viewed as adapting these ideas to RFs. The novelty lies in applying known second‑order techniques in a rectified flow context rather than in developing a fundamentally new algorithm.

- Heuristic design and lack of theoretical guarantees:  Both temporal and spatial approximations rely heavily on heuristics rather than theoretical justification. For instance, the temporal acceleration approximation in Eq. (6) becomes unreliable when delta t is large, which occurs in few-step sampling scenarios—likely explaining the underperformance of A²S in 4–5-step sampling in Table 7. Furthermore, the spatial acceleration compensation employs a fixed linear gain schedule (α_start = 0.95, α_end = 1.05), assuming linear dependence on spatial perturbations. Although some ablations compare different schedules, there is neither justification for this specific linear form nor analysis of stability or convergence. Empirical validation on only five samples is insufficient to substantiate the design choices. In addition, the paper does not discuss how the method interacts with adaptive step-size control or whether it risks overshooting in more complex prompts.

- Scope of baselines: The baselines reported in Table 1 are relatively outdated. Although the authors mention methods such as DPM++ and DEIS, these are not included in the comparison. Moreover, several recent samplers for flow matching models—such as FlowTurbo [1] and Differentiable Solver Search for Fast Diffusion Sampling [2]—are neither compared against nor discussed, limiting the completeness of the experimental evaluation.

---
[1] FlowTurbo: Towards Real-time Flow-Based Image Generation with Velocity Refiner \
[2] Differentiable Solver Search for Fast Diffusion Sampling

**Questions:**

- How does A²S compare to applying standard second-order ODE solvers, such as the classical Runge–Kutta or Heun’s method, directly to the rectified flow (RF) ODE? Would an adaptive Runge–Kutta method yield better performance? It would be valuable to include such comparisons under equal computational budgets.

- Many reported metrics (e.g., PIQA, AES, CLIP, HPSv2) are model-based. Have the authors conducted any human preference studies to verify whether the observed improvements are perceptible and align with human judgment?

- Why does A²S perform significantly worse than the baseline when using only 4–5 sampling steps? Please provide an explanation or analysis for this degradation.

- How many NFE steps does A²S require in Table 1, and what are the results without the caching algorithm (i.e., when using only the A²S sampling procedure described in Eq. 12)?

- Can you elaraborate about the implementation details of applying A2S to existing baselines such as DPM and UniPC?

---

### Official Review · Reviewer_NtBM · 2025-11-03

**Soundness:** 3
**Presentation:** 2
**Contribution:** 2
**Rating:** 4
**Confidence:** 5

**Summary:**

This paper proposes Acceleration-Aware Sampling (A2S), a lightweight and training-free method for improving few-step sampling in rectified flow (RF) models.
Aiming to address the curvature of the flow sampling ODE, this paper start with the second order expansion of the sampler and derived a time-shifted velocity and spatial scale modulation.
The method incurs no additional forward passes, making it model-agnostic and plug-and-play. Extensive experiments on FLUX-dev, Stable Diffusion 3.5, and several benchmarks demonstrate consistent quality and preference improvements at similar or lower latency.

**Strengths:**

1. The motivation and presentation of this work is very straightforward and easy to follow.
2. The method is plug-and-play, requires no retraining, and works across diverse pretrained models and datasets.
3. Strong empirical validation: The proposed method is evaluated on multiple datasets and evaluation metrics, and demonstrates consistent improvements in perceptual, aesthetic, and alignment metrics across benchmarks.
4. Extensive ablations show the effectiveness of temporal and spatial effects, and compatibility with existing samplers and distilled models.

**Weaknesses:**

1. Lack of in depth analysis and comparison to second order sampler methods. The authors are expected to show connections with existing higher-order samplers (midpoint/Heun), as A2S is an improvement on the first order Euler sampler.
2. Missing focus of evaluation. While A2S is proposed to tackle the "0 acceleration" issue **especially at few sampling steps**, the main experiment section focuses on 20 steps results (both tab1-4 and fig 2). Moreover, in fig 3b, base + A2S is worse than base when steps is less than 15; in Appendix D and tab 7, A2S show negative effects when the sampling steps further reduces to ~5, which hurt the initial assumption of the proposed methods. Actually, the use of $v_\theta(x_t,t+\frac{1}{2}\Delta t)$ when $\Delta t$ is large is ill-defined and OOD as the model is trained with almost perfect $(x_t,t)$ pairs.
3. Missing explanation in eq(11) second order term. In eq(10), the second order term should be $\kappa_t v_\theta(x_t,t)$, however, this becomes $\kappa_t v_\theta(x_t,t+\frac{1}{2}\Delta t)$. This change is not explained in the paper, like a sleight of hand. Without this approximation, two forward passes are required for one update.
4. As this method proposed as a clever and cheap approximation to avoid the JVP calculation (by using finite different, throwing away term, and approximation), it would be beneficial to see the full potential/uplimit of using eq(4) for sampling. The JVP implementation is not unacceptable, as demonstrated in recent work such as MeanFlow.
5. The fast samplers baselines (DPM, UniPC) are old and not designed for flow model (even flow and diffusion are same up to the scheduler), the authors should include also few new faster samplers as stronger baselines.

**Questions:**

See Weakness.

---

### Official Review · Reviewer_nxvL · 2025-11-11

**Soundness:** 2
**Presentation:** 2
**Contribution:** 2
**Rating:** 4
**Confidence:** 3

**Summary:**

The paper introduces Acceleration-Aware Sampling (A2S), an efficient sampling method without extra latency, for flow matching models (RF models). Most existing samplers, such as the first-order Euler method, assume constant velocity along the trajectory. The authors tackle this assumption, showing that the learned flow field actually changes over time and space. Thus, they argue that acceleration should also be considered during sampling and further decompose it into temporal and spatial components. Since explicitly computing acceleration is expensive, they introduce lightweight corrections, a) midpoint time-shifted velocity for temporal acceleration, b) scale scheduling for spatial acceleration. Empirically, across different datasets and backbones, A2S yields consistent improvements over existing sampling baselines.

**Strengths:**

1. Strong motivation: The idea of considering acceleration during sampling for flow-matching models is well-founded. Recent studies have started to explore second-order or acceleration-aware formulations for sampling or even during training, and this work aligns with these recent directions.
2. Conceptual clarity: Clear second-order framing for RF sampling error under coarse steps. The derivation and the decomposition of total acceleration into temporal and spatial components are intuitive, and well-explained (especially in the Fig 6 in the supplement).
3. Efficiency + versatility: A2S is efficient, since it has no additional latency. Also, it can be integrated easily as a plug-and-play component into existing samplers. The authors also demonstrate extra gain when combined with these samplers under the same budget.
4. Extensive experiments: using two different backbone models, comparing with multiple sampling baselines, and most of the results showing consistent (though a bit marginal) improvement.

**Weaknesses:**

Although I agree with the motivation and the overall conceptual framing, I'm not fully convinced by the current manuscript.

The paper presents itself as being “acceleration-aware,” but the implementation feels closer to a heuristic that stabilizes coarse-step Euler sampling through a combination of midpoint time-shift and step-size rescaling.
For temporal acceleration, the correction/compensation seems reasonable: using a 0.5 Δt time shift consistently improves results across different settings.


However, I'm not too sure about spatial acceleration, since it feels like a gap exists between theory and practice here.
The paper’s motivation and discussion (e.g., Fig. 1c, Sec. 3.4, ...) naturally give the impression that considering spatial acceleration would allow the sampler to account for or correct trajectory curvature and directional variation, yet the actual implementation simply rescales the step size to mitigate overshoot. This design choice feels conceptually detached from the stated goal of modeling spatial acceleration.
Empirically as well, the ablations (e.g., Tables 2 and 4) do not show clear benefits from the spatial acceleration.


Another (minor) point is,
It would be helpful to include ablation/analysis on the hyperparameters (e.g., δ and α) w.r.t. different NFEs. NFE versus performance gain could clarify whether the optimal hyperparameter settings differ between coarse-step and fine-step regimes.

**Questions:**

N/A

---

### Note · Authors · 2025-11-13

I have read and agree with the venue's withdrawal policy on behalf of myself and my co-authors.